# Beyond the 'dyad': a qualitative re-evaluation of the changing clinical consultation

Deborah Swinglehurst,[1] Celia Roberts,[2] Shuangyu Li,[3] Orest Weber,[4] Pascal Singy[4]

▶ Prepublication history and additional material paper is available. To view please visit the journal (http://dx.doi.org/10.1136/bmjopen-2014-006017).

[1]Centre for Primary Care and Public Health, Barts and The London School of Medicine and Dentistry, Queen Mary, University of London, London, UK
[2]Department of Education and Professional Studies, King's College London, London, UK
[3]Division of Medical Education, School of Medicine, King's College London, London, UK
[4]Psychiatric Liaison Service, Lausanne University Hospital, Lausanne, Switzerland

**Correspondence to**
Dr Deborah Swinglehurst;
d.swinglehurst@qmul.ac.uk

## ABSTRACT

**Objective:** To identify characteristics of consultations that do not conform to the traditionally understood communication 'dyad', in order to highlight implications for medical education and develop a reflective 'toolkit' for use by medical practitioners and educators in the analysis of consultations.

**Design:** A series of interdisciplinary research workshops spanning 12 months explored the social impact of globalisation and computerisation on the clinical consultation, focusing specifically on contemporary challenges to the clinician–patient dyad. Researchers presented detailed case studies of consultations, taken from their recent research projects. Drawing on concepts from applied sociolinguistics, further analysis of selected case studies prompted the identification of key emergent themes.

**Setting:** University departments in the UK and Switzerland.

**Participants:** Six researchers with backgrounds in medicine, applied linguistics, sociolinguistics and medical education. One workshop was also attended by PhD students conducting research on healthcare interactions.

**Results:** The contemporary consultation is characterised by a multiplicity of voices. Incorporation of additional voices in the consultation creates new forms of order (and *dis*order) in the interaction. The roles 'clinician' and 'patient' are blurred as they become increasingly distributed between different participants. These new consultation arrangements make new demands on clinicians, which lie beyond the scope of most educational programmes for clinical communication.

**Conclusions:** The consultation is changing. Traditional consultation models that assume a 'dyadic' consultation do not adequately incorporate the realities of many contemporary consultations. A paradox emerges between the need to manage consultations in a 'super-diverse' multilingual society, while also attending to increasing requirements for standardised protocol-driven approaches to care prompted by computer use. The tension between standardisation and flexibility requires addressing in educational contexts. Drawing on concepts from applied sociolinguistics and the findings of these research observations, the authors offer a reflective 'toolkit' of questions to ask of the consultation in the context of enquiry-based learning.

### Strengths and limitations of this study

- Brings insights from applied sociolinguistics to the analysis of consultations, including detailed interactional transcription and analytic concepts. These may be unfamiliar to some readers and we recognise it is not easy to make them accessible.
- Addresses the mismatch between consultations as conceptualised in communication models and the reality of many contemporary consultations.
- Offers a research-informed output, a 'reflective toolkit', for use in practice by clinicians and educators.
- Focuses on issues relevant to a globalised, technology-driven world, but does not address all types of consultation that breach the communication dyad (eg, clinician–patient–carer).

## INTRODUCTION

Two of the most significant changes affecting communication in the consultation are the increasing use of computers (the 'technologisation' of care)[1] and globalisation. The use of electronic patient records (EPRs) is gathering pace throughout Europe with the UK, the Netherlands and Scandinavia leading the way.[2–4] Globalisation (the movement of people, their languages, cultural practices, artefacts and 'norms' between countries) is creating 'super-diverse' multilingual populations.[5] According to the 2011 census in England and Wales, 29% of the population were born abroad or have a parent or grandparent born abroad. In Switzerland, 35.1% of those aged over 15 years are first-generation or second-generation migrants.[6] These social changes have significant impacts on the consultation.

Researchers of electronic patient records have coined the term 'triadic' consultation to highlight the computer as an influential third party in the consultation.[7–10] Swinglehurst et al[11 12] go further, conceptualising the EPR as bringing a wide range of competing voices

to the consultation and shaping its dynamics. Likewise, the dynamics of multilingual consultations are changed by the inclusion of professional and *ad hoc* interpreters (untrained family members, staff or volunteers). The resulting configuration has been referred to as a 'trialogue'.[13–18] An increasing number of consultations involve patients (and doctors) communicating in a language other than their first language, or in a variety of the majority language (eg, English) influenced by their first language. In these consultations the communication barrier may lead to a 'loss' of patient voice (examples 2 and 3 in this paper) or to unresolved misunderstandings arising from subtle differences in speech delivery, word stress and styles of self-presentation.[19]

Consultations that incorporate sociotechnical or sociolinguistic challenges (or both) are increasingly the norm in large urban areas. Although medical educators recognise that consultations are growing in complexity,[20] educational resources addressing these complexities remain limited. Current consultation models assume a communication 'dyad' in which two voices (patient and clinician) engage in focused interaction using broadly shared ways of communicating. Communication tends to be envisaged as a series of learned prototypical 'skills' or procedures for accomplishing clinical tasks rather than as a dynamic interaction that emerges moment-by-moment, shaped by every interactional nuance along the way. Assumptions about the nature of communication are reflected in strategies currently advocated for interpreted consultations such as: advising the interpreter on what is expected up front; explaining the interpreter's role to the patient; allowing ample time; asking one question at a time; clarifying confusing responses; and seeking 'cultural information' from the interpreter afterwards.[17 21–23] Likewise, in computer-mediated consultations, doctors are advised to avoid trying to attend to patient and computer at the same time (eg, by 'signposting' computer use), to use mobile monitors and to 'look at the patient'.[8 24] Although these suggestions are useful, they overlook the fact that the interaction is itself fundamentally and profoundly changed by these new arrangements.

This paper explores the characteristics of these contemporary consultations through presentation of case studies selected as 'telling cases',[25] highlighting the challenges arising in consultations that involve a meeting of more than two voices. Analytic observations are developed into a reflective 'toolkit' for use in the educational context while analysing learners' video-recorded consultations. For readers who may be unfamiliar with sociolinguistic concepts presented in this paper we include a list of definitions in box 1.

## METHODS

A series of interdisciplinary workshops was held over a 12-month period bringing together academics specialising in healthcare communication. Their disciplinary

---

| **Box 1** Definitions |
| --- |

**Voice**
Drawing on social theory 'voice' has both literal and metaphorical meanings. It is used literally as the human voice, that is, the sound of the voice and the manner in which someone speaks. 'Voice' is used metaphorically (1) in writing, to identify the distinctive style and authority that a text has, for example, the EPR (2) in speech and writing, as multiple or hybrid voices, when different styles are conflated together or a dominant style is infused with a less noticed one.

**Dyad and Triad**
'Dyad' is the traditional one-to-one communication between two people (here the clinician and patient), which is seen as the norm. A 'triad' is an interaction of three people or voices. Here the conventional two person communication is disturbed and its norms are challenged.

**Misalignment**
'Misalignments' are uncomfortable or inappropriate moments or instances where one side has difficulty interpreting the assumptions of the other. They are also moments when the speakers appear to be on parallel tracks, not responding fully to each other.

**Agent**
'Agent' is a term used in grammar analysis to describe the person/thing in the sentence, who/which is the main subject doing the action.

**Repair**
'Repair' is used metaphorically to describe how misunderstandings and misalignments in interaction are dealt with. It often involves talking *about* talk, to sort the interactional problem out.

**Social constructionism**
An approach which assumes that reality is the result of historical, social and political processes, in which the interest of the researcher is in how phenomena come into being, the processes by which they come to be 'constructed' as they are.

---

backgrounds spanned medicine, applied linguistics, sociolinguistics and medical education. Case study presentations were followed by discussion, leading to further analysis of primary interactional data. The case studies were selected from four ethnographic/sociolinguistic research projects drawing on theme-orientated discourse analysis,[26] conversation analysis[27] (CA) and linguistic ethnography.[28]

The selection of case studies was informed by case study methodology and based on a key ethnographic principle, that of 'developing theory through the study of critical cases' (page 20).[29] The workshops drew on Mitchell's concept of a 'telling' case study 'in which the particular circumstances surrounding a case serve to make previously obscure theoretical relationships apparent' (ref. 25, p.239).

So, telling cases from the four research projects were selected as examples of consultations that breach the clinician–patient 'dyad', incorporating additional 'voices'. The authors worked together to identify

synergies across them, teasing out themes and valid connections between events and phenomena relevant to medical education. Case studies explored the following: the interactional structure of general practitioners' (GPs) consultations involving interpreters; the shaping influence of the EPR in primary care consultations; and consultations involving patients communicating in a language other than their first language. The nature of the additional 'voices' in the latter example may not be immediately apparent, but relates to sociocultural scripts originating *beyond* the consultation and informing notions of how to present oneself, ideas about the self (and the clinician), one's relationship to authority and expectations of the healthcare system, for example. Although there is an extensive literature on cultural health beliefs, precisely how patients present themselves, how they voice their concerns and the impact of differences in linguistic background on the orderliness and distribution of knowledge and expertise have been much less studied.

## RESULTS

We identified two key inter-related themes, which are the main focus of this article: *orderliness* and *distribution*. We will begin by introducing these themes and will then present some short extracts of data analysis illustrating how these themes play out in the contemporary consultation and how they disrupt the dyadic nature of the consultation.

### 'Orderliness' in the contemporary consultation

The 'orderliness' of the consultation has been the subject of much previous research. Medical educators will be familiar with the stages of the consultation described by Byrne and Long,[30] and with more detailed models (eg, Calgary-Cambridge), which are currently favoured within educational curricula.[31] These models describe the consultation in more-or-less discrete phases such as 'gathering information' and 'explanation and planning'. Each stage is associated with a set of skills that underpin formative and summative assessments of medical students and some professional licensing examinations (eg, the UK Clinical Skills Assessment forms part of the licensing examination for GPs).

Apart from the assumption that the consultation, with its various 'phases', is an orderly affair, these models tend to assume a structuralist orientation to language, that is, the talk shared between clinician and patient is assumed to represent particular meanings—talk is simply *representative* of reality. For example, when a clinician 'summarises' the consultation, this summary is assumed to reflect a concise version of the patient's story, which is in turn assumed to represent the patient's experience. An alternative *social constructionist* perspective would also consider the additional work being *accomplished* by summarising—for example: the clinician's opportunity to take back the 'speaking floor'[32]; the

organisation of the story; the emphasis afforded to those aspects perceived to be most salient to diagnostic reasoning or clinical management; and the clinician's construction of their professional identity. From this perspective the encounter is relatively unstable, and the orderliness of the consultation, or its identified 'phases' are not so much inevitable *attributes* of the consultation, but are 'brought about' or 'worked up' through interaction. This 'bringing about' is informed by previous cumulative experience of what usually happens in the kind of interaction we recognise as a 'medical consultation', but involves a certain amount of improvisation along the way.

### 'Distribution' in the contemporary consultation

In consultations that lie beyond the 'dyad' by inclusion of additional people (eg, interpreters) or technologies (eg, electronic patient records) or patients whose first language is not English, we face new configurations in terms of the distribution of knowledge, power, authority and social identities. In what has been called the "crowded" consultation,[33] where many voices meet, new questions become salient and contested. For example: *Who is doing the talking? Whose voice is heard? How is knowledge distributed? What is important medical knowledge? Whose interests are being served? Who is the patient? Who is the clinician?*

### Analysis of 'orderliness' and 'distribution' of roles in the contemporary consultation

In this section we will illustrate how 'orderliness' and 'distribution' play out in consultations that breach the communication dyad, using selected data extracts. These examples combine the microanalytic methods used in CA with ethnographic observation of the relevant institutional contexts, mindful that this broader context shapes interaction in important ways. CA considers the detailed systematic patterns and regularities that arise as each speaker takes up, interprets and responds to the other's turn.[34 35] Our case studies show how the well-described orderliness of the consultation becomes disturbed when additional voices are introduced, as new forms of order and disorder emerge and care becomes increasingly distributed. We have retained the transcribing conventions used in each original study (see online supplementary appendix). The text is interspersed with suggestions of reflective questions that emerge from our data analysis. We anticipate that these questions will encourage tutors and learners to discover the importance of considering the consultation as an emergent *co-constructed* phenomenon, requiring a degree of improvisation. They are intended for use by tutors in undergraduate and postgraduate contexts (eg, GP training) when teaching clinical communication and also by learners as they play back and reflect on their own video-recorded consultations, sensitising them to particular challenges posed in these complex consultations and extending the range of available tools for critical analysis. Their value may be

**Figure 1** New forms of order and the distribution of authorship in the asthma clinic (EPR, electronic patient record; N, nurse; and P, patient).

| Time | N/P | Spoken word | Bodily conduct / notes on EPR |
|---|---|---|---|
| 01:08 | N | So really straightforward. | N puts paper on desk |
| | | (0.4) | N rotates body and gaze to face P, her hands on her lap. P looking at N |
| 01:09 | N | Asthma assessment | |
| | | (0.4) | |
| | P | Okay | P nods |
| 01:11 | N | to see how your asthma's do:ing: | N raises both hands in front |
| 01:13 | N | what you're doing w- with it when it's good, what you do with it when it's ba:d, (0.2) have you any problems with your ↑inhalers (0.4) .hhh | N uses fingers to count (on 'good', 'bad', 'problems') |
| | | (0.5) | N hands open out in front of her |
| 01:19 | N | Very straightforward stuff | N hands to lap |
| | P | Oka[y | P nods |
| | N | [all right? .hhh | |
| {lines of transcript omitted} | | | |
| 02:46 | N | So::: er (( C )) | N turns to face EPR screen |
| 02:47 | N | Ho- you've never smoked. | N → EPR (last entry in template reads 'never smoked') |
| | | (0.4) | |
| | N | that's what [I've got here | N turns head towards P |
| | P | [no (0.4) [never smoked] | P looking at N |
| 02:50 | N | [never smoked] (( C )) [Exc] ell [ent] [(( C ))] [(( C ))] | N turns back to look at EPR. Emphatic keystroke |
| 02:52 | N | [Th] at's gr:[e::at ] [(( C))] [(( C ))] | N looking at screen, typing keystrokes |

enhanced by educational support from a sociolinguist or a more collaborative interdisciplinary pedagogy that brings together clinicians and linguists.

First, we look at the opening of a consultation between a nurse and patient, English speakers who have not met before (figure 1). The institutional context is an annual asthma check, a requirement of the UK Quality and Outcomes Framework (QOF) for which incentive payments are made. The nurse is completing a computer template (form) during the consultation. The transcript includes notes on bodily conduct and the computer screen display.

She frames the consultation as an 'assessment' emphasising it is really (01:08) or very (01:19) straightforward. The linearity of the upcoming consultation is alluded to as she counts a three-part list with her fingers. She demarcates the purpose of the clinic, laying it out as an orderly affair and (implicitly) setting limits on what can happen. She adds to this later (at 2:09, transcript not shown), while gesturing towards the EPR: *"What I've got here is some questions that I—I need to ask you. They're fairly straightforward ones but what they tend to do with is that they will flag up whether there >actually< we have got what w- what I would call breakthrough symptoms."* Reiterating that it is

'straightforward', the phrase *"I need to ask you"* points to an underlying institutional requirement. Reassurance focuses on an anticipated orderliness of the clinic, dealing up front with any misalignment between what the patient may expect and what the nurse is required to do. However, this is a different kind of order to that which we might expect. This is an orderliness in which the electronic template is instrumental, rather than one which emerges through dialogue between clinician and patient.[12]

The nurse then goes on to speak *not* of symptoms, but of inhalers (transcript omitted) and then smoking. Here the template introduces a topic (smoking), which seems tangential to this patient's particular circumstances, although it is important to asthma care in general terms, and has institutional significance, being a QOF indicator. The EPR thus brings an institutional voice into the encounter, making relevant the patient's identity as a lifelong non-smoker in this context. It contributes to defining what is important medical knowledge, reproducing particular definitions of 'quality' in practice—gathering data about (non-)smoking for QOF being an example. The patient becomes an epidemiological informant and 'quality' is transformed into meeting an

> **Box 2** Reflective questions based on case 1, which might inform analysis of a student's own video-recorded consultations
>
> Which 'voices' can I identify as being present in this consultation?
>   Which voices are being privileged at different times in the consultation and why?
>   What is the consequence of this?
>   How do I ensure that the patient's voice is not lost?
>
> How and to what extent do I need to reshape my own communication norms/style to accommodate the specific arrangement of people and computer in this consultation?
>
> How and to what extent am I fully 'involved' in this consultation?
>   What does this mean to me and what challenge is this particular consultation presenting?
>   To whom and to what am I attending, and with what purpose?
>
> How am I incorporating computer templates and prompts?
>   What is the consequence of my communication with the patient?
>   To what extent is the sequencing and ordering of our talk being influenced, if at all, by the demands of the EPR?
>   Do I need to consider possible alternative ways of managing this situation?
>
> How does interacting with the computer affect the standard models of good communication in the textbooks?

institutional requirement rather than focusing on the specific quality of care of the individual. At 2:50–2:52 the nurse's emphatic evaluations "*excellent, that's great*" are spoken towards the EPR as she types, apparently referring to her satisfaction at meeting its demands, the patient watching from the sidelines. At the very least it is ambiguous to whom (or what) these superlatives relate.

The analytic question of "*Who is doing the talking?*" is at issue here. For example, the authorship of the words at 2:47 is apparently distributed between the nurse and the EPR. We see a disruption of the usual conventions of conversation, with this comment spoken by the nurse as she looks at the computer screen rather than at the patient to whom she expects to hand over the speaking 'turn'. The nurse's attention is divided between what has been called the patient *embodied* and the patient *inscribed*[36] [37] as the patient becomes (metaphorically) distributed between person and record.[36] The nurse is *cognitively* oriented to the patient as she establishes his

smoking status by looking at the EPR, but the *affective* aspect of her involvement—which Goffman[38] has highlighted as crucial in a social interaction—is compromised. This 'template talk' is met by a 0.4 s pause. The nurse then turns to face the patient, adding "*that's what I've got here*"—this time evoking a response, as they jointly repeat "*never smoked*", words that were initially displayed on the EPR screen (visible only to the nurse).

Box 2 suggests some reflective questions to ask of consultations involving the EPR. Students may find it helpful to use the transcript in figure 1 to gain familiarity with these questions before asking the same questions of their own video-recorded consultations. In the following sections, we incorporate reflective questions arising from our data analysis as applicable to different kinds of complex consultations.

Our next example (figure 2) is from a GP consultation with a patient from Nigeria who speaks a variety of English that differs from local English. While on holiday

**Figure 2** Disorder arising from different conventions in intonation (GP, general practitioner; PT, patient).

**Dis-order arising from different conventions in intonation**

| 1 | GP | what kind of dog was that (.) it was somebody's (.) dog= |
|---|---|---|
| 2 | Pt | =yes somebody's |
| 3 | GP | it was a stray dog |
| 4 | Pt | no no it was somebody's dog |
| 5 | GP | right |
| 6 | Pt | yes I:: made an enquiry they said that- **they *they told me** |
| 7 | | **the dog go to the vet regular** |
| 8 | GP | *right *okay |
| 9 | Pt | **but that's *what they said** |
| 10 | GP | right (.) *right right so did you know the owner or did= |
| 11 | Pt | = I know the owner = |
| 12 | GP | = oh fair enough (.) so |
| 13 | Pt | **erm:: ((laughs)) but** |
| 14 | GP | did you see any doctor then |
| 15 | Pt | no |

in Nigeria the patient was bitten by a dog, raising the question of whether he might have been exposed to rabies (lines 1–4) and considered for vaccination.

This example illustrates an unresolved ambiguity that arises from different conventions in standard English and Nigerian English over the use of contrastive stress (indicated by * in data). Disorder, misunderstanding and an incorrect assessment ensue. In lines 4–7 the patient tells the GP he knows the dog's owner and that the dog visits the veterinary surgeon regularly. This reassures the GP that the patient is at low risk (line 12: "*oh fair enough*") and he in turn later reassures the patient that no vaccination is needed. In fact the patient was conveying his concern that the dog may **not** be free of rabies, and that the owner could not be trusted. When the patient says "*they told me the dog go to the vet regular but that's <u>what</u> they said*" (lines 6,7,9, highlighted in bold in

transcript) he emphases the agent ('*they*') and the content of the agent's talk ("*what*"). The equivalent sentiment in standard English would be "*they <u>told</u> me the dog goes to the vet regular, but that's what they <u>said</u>*" with the emphasis on the verbs ("told", "said"). Although the patient offers further hints of his scepticism in line 13— a hesitation, laughter and the word "but"—the underlying ambiguity is passed over, and absent from the institutional record made by the doctor. Misunderstandings, or the illusion of understanding, which result from small and subliminally processed differences in talk, are more frequent in a multilingual patient population[19] and challenge conventional orderliness. However, no simple behaviours can be taught in situations of super-diversity as our next two examples also show.

Figure 3 shows an extract of a video-recorded consultation in a Swiss pain clinic (translated from French) in

**Negotiating consent for a spinal injection**

103. Dr: because I thought I'd give you an injection today
104. Pt: =no
105. Dr: =yes, give one
106. Pt: =yes
107. Dr: But if you don't want, I won't. I, for me, it's not a problem giving you an injection
108. Pt: =Yeah
109. Dr: =I don't think that it's a big problem for you either
110. Pt: mmh mmh.
111. Dr: but I think that after the other injection, you had a benefit, you felt better after the injection in October . . Didn't you?
112. Pt: mmh. ((*mimics express perplexity*))
113. Dr: So that's why I'd like to do it again, if you, if you don't want, no, I won't do it, if you say, er, you are afraid, you say, er, after you are getting paralysed or whatever, I won't do it.
114. Pt: mmh . well it's me now not think more, not think right . . not er er . . er don't know it's the cause that how is paralysed, for me speaking?
115. Dr: =you [told me]
116. Pt:      [oh but yes]
117. Dr: you told me before
118. Pt: =yeah
119. Dr: =you said, you were afraid of getting paralysed
120. Pt: =yeah
121. Dr: =so, this happens not often
122. Pt: =and next, the next (1) here keep . water. ((*patient indicates non verbally something running down from his forehead to his temples*))
123. Dr: =where?
124. Pt: here, next (1) er injection, it's me, it's the, it's the, back pain, me come, immediately injection ah . . ah, that's not much pain, after. problem, it's same like war, ah . in my country, war and then, I don't know how to explain well
125. Dr: well, I don't know, if, today, I tell you we give the injection, we're gonna give the injection, do you agree or not
126. Pt: =yes
127. Dr: =then we'll give it

1. In this translation "next" probably means "last" – this was a recurring error of vocabulary throughout the consultation

**Figure 3** Negotiating consent for a spinal injection (Dr, doctor; Pt, patient).

**Box 3** Reflective questions based on cases 2 and 3, which might inform analysis of a student's own video-recorded consultations

Which 'voices' can I identify as being present in this consultation?

How do I need to adjust my approach to the consultation when the talk *itself* seems to be the problem?

Am I confident that I correctly understood the patient's problem, in the knowledge that subtle features such as word stress and styles of self-presentation might differ in speakers whose variety of English is influenced by a language other than my own? If not, what were the other possible meanings of this section of talk?

How can I ensure that I clarify the patient's intended meanings?

Did the strategies that I used to do relational work in the consultation have the desired effect in this multilingual consultation? (*Examples might include the use of humour, metaphor, or attempts at 'informal' conversational styles.*) Did I correctly identify the patient's attempts at relational work?

This consultation felt muddled and chaotic and did not evolve as I was expecting. Why might this be? Does my explanation reveal any underlying assumptions about how I understand the act of consulting, my expectations for the consultation and my role as the clinician?

At what point do I decide I cannot consult effectively without an interpreter either because it is not clear whether the patient and I have understood each other or because I am concerned that the patient's voice is being lost?

Do the models of patient-centredness and patient-shared decision-making work when talk itself seems to be the problem?

which consent is sought for a spinal injection. The patient left his home country 10 years earlier after a war, and has very limited command of French. He has chronic low back pain for which he has received spinal injections. He is worried about the risks involved, because a previous consent form (translated by his daughter) referred to a risk of paralysis.

As in the previous example, there is obvious asymmetry of the linguistic resources available. Indeed, the entire extract might be regarded as a continuous, extended and only partly repaired misunderstanding. Both parties struggle to grasp at least some meaning in the words of the other. An interview with both interactants afterwards (data not shown) suggested that some areas of shared understanding were reached: the patient's fear of the injection, the positive impact of a previous injection and agreement to proceed today. However other communicative efforts failed, including the doctor's attempt to reassure the patient that paralysis does not happen often. Indeed, the patient reported that the doctor had particularly reminded him of the risks of paralysis adding that he, the patient, had to take responsibility for this risk. The patient's fear (linked, it appears, to some past experience in the war) is not explored. Arguably, under conditions of such scarce common linguistic means this topic may be deemed too complex to tackle; the patient's voice from the past is lost.

Difficulties arising in the clinical decision-making process have previously been described in monolingual contexts,[39 40] but they become magnified when patients have limited proficiency in the language of consultation. In this multilingual context the ideals of shared decision-making within an ordered consultation are eroded. In this example, the doctor first presents his idea, tries to convince the patient of the benefits of another injection and goes on to investigate the patient's worries and possible reasons for disagreement. However, the participants fail to connect the discursive threads of this discussion to the final decision. The patient consent follows immediately after major interactional troubles, culminating in a self-critical metacommunicative account by the patient ("*I don't know how to explain well*") and an abrupt topic shift by the doctor. The doctor appears to cut to the decision-making when he gives up on achieving further clarity about the patient's stance.

Box 3 presents some reflective questions to ask of consultations that involve patients who have either a limited command of the dominant language of the consultation, or speak a non-local variety of the dominant language (such as those illustrated in figures 2 and 3). These questions are intended to inform analysis of students' own video-recorded consultations.

Consultations such as those in figures 2 and 3, in which the clinician and patient are not 'in tune' with each other require considerable and not always successful collaborative work by both parties to prevent, recognise and repair misunderstandings.[41] The use of interpreters may address these issues to some extent, but not without introducing different challenges. The work of consulting becomes distributed between at least three participants, changing the relationship of the speakers to their own words and so disturbing roles and identities. The orderliness in interpreted consultations is changed, both in terms of overall structure and in microinteractional patterns. Clinicians have to do more than simply establish consensus on the mode of communication and the role of the interpreter as suggested by recent guidelines.[16 17] At the microlevel, extra verbal exchanges are required to clarify misunderstandings (as we saw in figures 2 and 3). The traditional doctor↔patient (dyadic) interactional sequence becomes a more

**Figure 4** Lost voices in distributed turn-taking sequence organisation (English translation in italics; Dr, doctor; Int, interpreter; Pt, patient).

**Lost voices in distributed turn-taking sequence organisation (English translation in italics)**

1.  Dr:      =An (.) she says it starts in the afternoon::n (.) every day (0.3) does it last (0.4) until
2.  she goes to bed, how long does it last for °in the afternoon°
3.  Int:      ako dlho to trvá
4.          *How long does it last for*
5.  Pt:       tak je to hodinu, dva, to strašne bolí a potom prestane.
6.          *well, it is for an hour, two, it hurts badly and then it stops.*
7.  Int:      ok, that start afternoon. She feel that pain about for one two hours
8.  Dr:       one or two h[ours
9.  Int:                  [and after that go away:: °ok°
10. Dr:  alright↑. Interesting. That's good.
11. Pt:  A ešte by som sa chcela povedať, [že ja nosím oku okuliare.
12.      *And I would also like to say that I wear gla glasses.*
13. Dr:                          [could you (0.59) show me where she feels it?
14. Int:  u:hm Môžte ukáza[ť
15.      *U:hm Can you show*
16. Pt:                  [Tu.
17.                  *Here*

complicated 'triadic' pattern. Close inspection of interpreted consultations shows that this assumed prototypical triadic sequence (doctor→interpreter→patient or patient→interpreter→doctor) is not always observed by participants, so that one or more participants' voices are 'lost'.[16 42] The power to decide who talks next is *unequally* distributed among the three participants—with patients at the bottom of the hierarchy. Even when the prototypical doctor–interpreter–patient sequence *is* followed, there remains considerable scope for misunderstanding, due to ambiguity over the interpreter's role and how the interpreting task is actually performed in practice. The interpreter delivers a 'hybrid' voice, which incorporates the voices of all three participants in the interaction. Figure 4 shows an extract from an interpreted consultation in England with a Czech-speaking patient who has reported of headache. It gives some insight into how the themes of orderliness and distribution play out in an interpreted consultation. A related set of reflective questions that may be useful to students as they analyse their own interpreter-mediated consultations is provided in box 4.

From lines 1–10, the participants follow the triadic sequence in their turn-taking. However, what the doctor and patient hear is reformulated by the interpreter. In line 1, the doctor asks a question. Prior to the question, he refers to an earlier discussion about the patient's pain, signposting a change of topic. However, this signposting is omitted by the interpreter in lines 3–4. Similarly, in lines 5–7 the interpreter adds '*that start afternoon*' into her translation although the patient did not say these words here. As Bolden points out, an interpreter is constantly choosing the quantity and quality of information that is translated, thus creating a hybrid voice and assigning themselves an extra role as either 'doctor' or 'patient' with blurring of the usual boundaries between the two.[43] The potential for voices to become 'lost' in this process is greatest when the prototypical triadic sequence is not followed.

At line 10, the doctor comments on the interpreter's response ("*Interesting. That's good*"). If the (assumed) triadic sequence was followed, one would expect either that the doctor would continue talking at this point, or that he would pass the speaking turn to the interpreter

---

| **Box 4**   Reflective questions based on case 4, which might inform analysis of a student's own video-recorded consultations |
|---|
| Which 'voices' can I identify as being present in this consultation? |
| How and to what extent do I need to reshape my own communication norms/style to accommodate the specific arrangement of people in this consultation? Do the models of communication in the textbooks need to be adapted in this situation? |
| How confident am I that this interpreter is doing what he or she is supposed to do? |
| When do I notice that the sequence of speakers (doctor, interpreter, patient) is different from that which I might expect?<br>What may have been the consequences of this disruption to the order of speakers on the understandings of this consultation?<br>  Do I notice occasions when the patient's voice is lost, that is, words of the patient appear to have gone without translation by the interpreter?<br>  Do I notice occasions when my own words appear to have gone without translation by the interpreter? |
| What can I do to ensure that the interpreter is working to the mutual benefit of the patient and doctor? |

to translate his words. Either way, we would not expect the patient to have a turn here. However, the patient (line 11) brings in a new topic, ostensibly in an 'inappropriate' place. Her utterance is not translated and goes unheard as the doctor interrupts the patient (line 13) before she can finish her talk; this marks the patient's entry as 'not legitimate'. The doctor and the interpreter then continue the conversation following the triadic sequence. In the wider data set from which this extract is taken, we found that doctors also speak at such 'inappropriate places'. However, there is a difference. In most cases, the words of the doctor *are* translated, and the patient is put 'on hold' while this is performed. In other words, when the prototypical doctor–interpreter–patient sequence breaks down it is the *interpreter* who takes on the role of 'distributor' of speaking turns and decides whose voice will be preferentially heard. In our data set, interpreters tended to prioritise the doctor's right to speak, as illustrated in figure 4.

## DISCUSSION

Rapid technological and demographic change has brought challenges to the consultation, which were not anticipated when the consultation models currently taught to students were developed. Using a selection of 'telling' cases as a basis for analysis we have been able to develop novel conceptual ideas about the contemporary consultation, which challenge normative assumptions, showing that the notion of the consultation as a dyadic meeting of two speakers who share communicative resources is frequently challenged. Our priority has been on depth of analysis rather than breadth, with our selection of case studies informed by the 'opportunity to learn' rather than by concerns around 'typicality'.[44] Based on a detailed study of four contrasting cases we suggest there are complex new configurations of voices in the consultation, and—as a consequence of this—the potential for 'losing' the patient voice. These challenges to the 'dyadic' consultation rarely receive explicit attention in the educational curriculum. One striking observation that emerges from our data is that the twin social pressures of globalisation and technologisation appear to place paradoxically opposing demands on the consultation. On the one hand, clinicians are challenged with increasingly diverse, unpredictable consultations from a sociolinguistic perspective, requiring flexibility and a tolerance of ambiguity. On the other, there is increasing pressure to 'standardise' practices, for example, through greater use of EPR templates.

Building on our analysis of these case studies we have offered a series of reflective questions that may be relevant to ask of complex consultations that take on these new kinds of orderliness and in which conventional understandings of the roles of clinician and patient become blurred. These questions have not yet been tested empirically in an educational setting and do not constitute a definitive checklist. They may neither be relevant to all consultations nor necessarily comprehensive, but we hope that they are a starting point to promote observation and discussion about the consultation from an orientation that embraces its new complexities. Further empirical research is required to test the value of this toolkit as an educational intervention in practice and to refine it in the context of further educational research.

We would like to invite debate among medical educators about how to adapt, extend or revise consultation models to ensure that these important aspects of the contemporary consultation do not remain overlooked. We suggest that an orientation to the consultation as a dynamic process that is co-constructed between clinician and patient is helpful, one in which the structure (we prefer 'orderliness') emerges out of the collaborative work of clinician and patient (and others) and which depends on how the 'work' of consulting is distributed between participants. Regarding the consultation as a co-construction demands more than a range of 'add-on' prompts describing specific clinician behaviours. It encompasses a shift away from the idea that consulting is a set of competences to be mastered, towards a more analytical orientation. The most important overarching question to ask of the consultation shifts from "*Did I do that well?*" towards "*What did we accomplish there?*" This brings the contribution of the patient and all relevant parties (or 'voices') into clearer view. The questions we offer to learners within our reflective 'toolkit' in this paper fall broadly within this overarching question.

We urge educators to consider critically how their approach to teaching clinical communication might change if instead of assuming that the talk *represents* some kind of existing reality they also encourage students to consider conceptualising talk as *constructing* reality, an assumption that underpins this paper. We suggest that greater use of the detailed analysis of video recordings of real (as opposed to simulated) consultations may be helpful, exposing learners—as consulters and critical observers—to the kinds of complexities that our research highlights. For example, a DVD by Roberts *et al* entitled "*Doing the Lambeth Talk*" shows how misunderstandings in the multilingual consultation can be avoided and repaired.

**Acknowledgements** The authors would like to thank the staff and patients who agreed to take part in the four original research projects from which the data presented in this article are selected. They would also like to thank their peer reviewers for helpful comments on the manuscript.

**Contributors** DS and CR took responsibility for developing the analysis into practitioner-relevant resources, refining these in discussion with SL, OW and PS. DS wrote the first draft of the paper and revised it in response to critical commentary from all of the remaining authors. All authors have approved the final version of the manuscript.

**Funding** DS received grants from the National Institute for Health Research Doctoral Fellowship and Medical Research Council (MRC). CR received grants from the Sir Siegmund Warburg Voluntary Settlement. SL received PhD Fellowship from the NHS Bradford & Airedale. OW and PS received grants from the Swiss National Science Foundation.

**Competing interests** None.

**Ethics approval** This study of itself did not require ethics approval but draws on previous research projects which were given ethical approval: Thames Valley multicentre REC 06/MRE12/81; NHS Bradford REC 09/H1302/106; REC of Vaud, Switzerland 271/07; and St. Thomas' Hospital local REC.

**Provenance and peer review** Not commissioned; externally peer reviewed.

**Data sharing statement** No additional data are available.

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
