## [Reviewer comments · BMJ Open]

This paper was submitted to the JECH but declined for publication following peer review. The authors addressed the reviewers' comments and submitted the revised paper to BMJ Open. The paper was subsequently accepted for publication at BMJ Open.

ARTICLE DETAILS

TITLE (PROVISIONAL)	Beyond the 'dyad': A qualitative re-evaluation of the changing clinical consultation
AUTHORS	Swinglehurst, Deborah ; Roberts, Celia; Li, Shuangyu; Weber, Orest; Singy, Pascal

VERSION 1 - REVIEW

REVIEWER	Fiona Stevenson University College London, UK
REVIEW RETURNED	23-Jul-2014

GENERAL COMMENTS	This is an interesting and timely addition to the literature on communication in clinical encounters. It critically considers the notion of a 'dyadic' consultation in the light of contemporary medical practice, namely the contradictory pressures of multilingual societies which require greater flexibility in consulting practices, together with the increasing pressure to 'standardise' practice. The article also presents details of a reflexive 'toolkit' of questions which may be used by clinicians and educators to reflect on consultation practice. The analysis presented is both interesting and thoughtful and provides an excellent contribution to the understanding of clinical decision making in modern society. I was however slightly unclear about the ways in which the authors envisaged the toolkits provided being used. The content of the toolkits are well grounded in the data presented and would prove a useful tool for reflection on consultations, however I would have thought that their use would require a supportive education environment and would have liked to have seen some indication of the types of settings to which they thought they would be best fitted. This applies particularly to some of the aspects in box 2, for example the idea of word stress and styles of self-presentation. Some reflections on how the debates raised here could be initiated with medical educators would also be helpful. Overall this paper is a well written and thought provoking addition to the literature on communication in clinical encounters.
---

REVIEWER	Dr Marie Manidis University of Technology Sydney Australia
REVIEW RETURNED	24-Jul-2014

GENERAL COMMENTS	Thank you for the opportunity of reviewing this paper. The authors
--

have tackled educational and communication issues relating to GP consultations in a multidisciplinary way using multi study data. I found this to be refreshing and useful.

Addressing the growing complexity of healthcare consultations is timely (if not overdue) and the authors have identified a crucial paradox, i.e. of doctors and nurses being under increasing pressure to standardise through the use of EPR technologies, yet needing to be simultaneously more flexible (and non standardised) in their communication with diverse patients. Maybe more could have been made of this tension earlier on in the paper.

I think that the point made about seeing consultation as emergent and unpredictable which shifts communication education away from teaching 'simple behaviours' is fundamental. The authors support this contention by suggesting that medical educators (and learners) look more analytically at what is going in this dynamic interaction and focus less on teaching formulae and scripts which have dominated for so long.

There may be one or two ways that the authors might strengthen the paper. Firstly, I had initial concerns with the way that the linguistic data were presented in their original (but different) forms from each of the studies. For medical and nursing readers, who might not be used to reading linguistic/and or conversational data, this could be confusing as the different conventions emphasise different analyses. However this point is merely raised as a one which could be addressed in the study's limitations (which have not been sufficiently addressed).

Secondly, the authors use the term 'voice' as a central concept in the paper. Could I suggest that for non linguistic readers, they define this term? It refers in different places to a speaker, to another person present in the consultation (i.e. the interpreter) and also to the 'voice' of the computer or the institution. On yet another occasion, it is used as a verb (page 8). In doing this, it straddles literal and metaphoric meanings which may be confusing for some readers. For those not familiar with linguistic terminology, this word could be defined in the beginning when first used. Some medical educators however may be familiar with the term.

In addition, the authors refer to 'learners' (and on another occasion they use the word 'student' (page 12)). It might be useful to make it a little clearer who the learners are - i.e. the clinicians who watch their own video recorded consultations.

The term 'discursive dispersion' which appears on page 15, line 9, is also introduced with little explanation.

One typographical error is noted on page 13, the word consultation may require an 's' - just before the insert box 1 caption.

I think the paper engages with the contemporary complexity of consultations in a useful way. In doing so it has introduced a number of theoretical approaches and lenses to the consultation, i.e. case study methodology, 'telling case study', theme-oriented discourse analysis, linguistic ethnography, social constructionist perspectives, Conversation Analysis, ethnographic observation etc. In addition many linguistic terms like 'text', 'dyad', 'voice', 'interactional structure' etc. I do not know how the plethora of terms and approaches might

	be simplified - maybe this is not needed. But I would encourage the authors to consider what this might mean for some readers.
--	--

REVIEWER	Geraldine Leydon University of Southampton UK
REVIEW RETURNED	01-Aug-2014

GENERAL COMMENTS	Methods: it is unclear how many workshops were run ('a series' is mentioned). It is also unclear how 'data' were captured at the workshops, how participants were selected/why? Why a workshop approach was felt to be appropriate/best given the microanalytic nature of the data then used to build their argument. There seemed to be a bit of a disconnect between the discussion and the research / findings reported. The discussion is very much about larger concepts of globalisation and use of technologies (such as EPR). The results begin to deal with what I think are some important and interesting issues but are not specified closely enough and do not link clearly enough with the larger concepts discussed in the Discussion. I would prefer to see a focus on one type of consultation rather than include examples of interpreted consultations, when English is not the patient's first language, and when EPRs are used. Each is tackled in what seems to be a brief way. The authors could by contrast deal in a lot of detail with a smaller type of problem/phenomenon e.g. interpreted consultations and the way in which interpreters privilege the voice of medicine/the GP/Nurse over the voice of the patient. What devices do they use, how do they accomplish this, what are the consequences? And the authors could discuss/show how patients (and GPs) resist/negotiate interpreter control. Or a paper that focuses on the use of computer/electronic patient records and the impact on interaction (although video data would probably be preferable for such an analysis). I ended up feeling a little cheated because the analysis just didn't get into a single problem in enough depth. It would help to focus the analysis and have more cases showing a single phenomenon. I also wondered whether we need to see a quotation from a seminal training text which illustrates the authors' critique that conventional wisdom in communication 'skills' training is to characterise the consultation as a dyadic encounter. The authors are critical of standard / established approaches and it would be good if they evidence their critique more so; it might also help to set up the contrast between how communication is construed in texts/teaching vs. how communication plays out in real (not simulated) settings. I really like some of the ideas in the paper. There are multiple voices in the consultation and empirical work such as that reported is excellently placed to begin to detail whose voices, the work they can do and the consequences they can have. Equally, I agree with the authors that there is a need to use real consultation data in educational settings/in communication training. Simulated interactions cannot match the (authentic) insights real time interactional data can provide (that's not to say there is not a role for simulated work). Moreover, I am aware that some of the authors have great expertise at using real data and are well placed to tackle
---

	a topic and analysis such as that addressed in the submitted manuscript. Unfortunately, however, I cannot recommend the manuscript for publication as it is currently configured. The main reason, as indicated above, is analytically I think the paper needs developing. The authors could either provide more evidence of how globalisation/technology impacts on interaction in these encounters - since these are the larger concepts they are dealing with. Or the authors could, as suggested above, re-focus the paper to look in more detail at one particular type of consultation e.g. the interpreted consultation. I hope the authors will feel they can revise the paper and resubmit because there are some interesting and important issues raised that warrant attention/discussion/debate. It might be the authors feel they can refocus the paper fairly easily which might constitute a minor rather than a major revision. There is a bit of jargon which could be simplified or explained. e.g. 'repaired misunderstanding' (repair is not something the BMJ Open audience will necessarily know about) p14 e.g. 'discursive dispersion/fragmentation' p15
--	--

VERSION 1 – AUTHOR RESPONSE

The third reviewer, Geraldine Leydon identified a number of strengths and interesting concepts in the paper but has suggested a much more radical overhaul (arguably a very different paper) specifically suggesting a more limited focus on one type of consultation (e.g. interpreter-mediated consultations or EPR-mediated consultations). Given the Editorial decision to make some minor amendments pre-publication, we are not in a position to respond to this reviewer’s comments in detail without re-crafting the manuscript along very different lines into what we suspect would be a paper serving rather different ends to those intended in this manuscript. We would also like to point out that the authors have already published on their work individually (references can be found in the reference list and include, for example, a paper by Li which discusses in detail some of the points Leydon raises on interpreted consultations and also two papers by Swinglehurst describing a linguistic ethnographic study based on video-recordings of EPR-mediated consultations). In these published papers the authors take as their focus one particular kind of consultation as Leydon suggests. However the main emphasis of this manuscript under review is a consideration of what these studies collectively reveal about the normative concept of the consultation as a ‘dyad’ and what might be the implications - an emphasis which very deliberately cuts across the four studies on which we draw. We are grateful to Geraldine Leydon for highlighting the need to deal with some of the ‘jargon’ which may be unfamiliar to readers of BMJOpen, a point also made by Dr Marie Manidis. We concluded that the best way of dealing with this was to include an additional box in which we explain the more unfamiliar terms.

We will respond to the first two reviewers points one by one.

Fiona Stevenson

1) I was however slightly unclear about the ways in which the authors envisaged the toolkits provided being used. The content of the toolkits are well grounded in the data presented and would prove a useful tool for reflection on consultations, however I would have thought that their use would require a supportive education environment and would have liked to have seen some indication of the types of settings to which they thought they would be best fitted. This applies particularly to some of the aspects in box 2, for example the idea of word stress and styles of self-presentation. Some reflections on how the debates raised here could be initiated with medical educators would also be helpful.

Thank you for this suggestion. We have incorporated some additional detail about possible contexts for this teaching on pages 10-11, and have highlighted the importance of appropriate educational

support, in particular suggesting that an approach which brings together teachers with backgrounds in clinical medicine and sociolinguistics. We have kept this fairly brief, as these are suggestions and not (yet) based on robust evaluation of these arrangements in educational practice. Regarding the extent to which this paper is capable of prompting debate...to some extent we feel that its publication represents a start in this process, and part of the reason for publishing this is of course to invite such debate.

Dr Marie Manidis

1) Addressing the growing complexity of healthcare consultations is timely (if not overdue) and the authors have identified a crucial paradox, i.e. of doctors and nurses being under increasing pressure to standardise through the use of EPR technologies, yet needing to be simultaneously more flexible (and non standardised) in their communication with diverse patients. Maybe more could have been made of this tension earlier on in the paper.

We have made a very minor amendment with this in mind by adding a sentence into our abstract to prepare people for this finding.

2) Firstly, I had initial concerns with the way that the linguistic data were presented in their original (but different) forms from each of the studies. For medical and nursing readers, who might not be used to reading linguistic/and or conversational data, this could be confusing as the different conventions emphasise different analyses. However this point is merely raised as a one which could be addressed in the study's limitations (which have not been sufficiently addressed).

We are aware of this and it is something the authors discussed at some length before submitting the paper for consideration of publication. We felt uncomfortable about changing original transcriptions (for obvious reasons) and when we considered it in more detail we realised that with the exception of a different way of indicating emphasis (underline or *) the differences were mostly presentational, all of the studies drawing broadly on the transcription conventions of CA for transcribing talk. Figure 1 (EPR study) clearly includes more attention to the bodily conduct of interactants, but as you point out this is a difference in analytic orientation and to alter this would probably introduce more problems than it solves. We have made a brief reference to this shortcoming in the article summary.

3) the authors use the term 'voice' as a central concept in the paper. Could I suggest that for non linguistic readers, they define this term? It refers in different places to a speaker, to another person present in the consultation (i.e. the interpreter) and also to the 'voice' of the computer or the institution. On yet another occasion, it is used as a verb (page 8). In doing this, it straddles literal and metaphoric meanings which may be confusing for some readers. For those not familiar with linguistic terminology, this word could be defined in the beginning when first used. Some medical educators however may be familiar with the term.

We realise this is not the only term that may be unfamiliar although it is of central importance. Reviewer 3 also pointed to the issue that non-linguists might find the terminology in the paper difficult in places. We have identified this as a limitation, and respond to it by including a box of definitions (which we have kept deliberately as simple as possible for the non-expert reader)

4) In addition, the authors refer to 'learners' (and on another occasion they use the word 'student' (page 12)). It might be useful to make it a little clearer who the learners are - i.e. the clinicians who watch their own video recorded consultations.

We hope the additional material in pages 10 - 11 now makes this clearer.

5) The term "discursive dispersion" which appears on page 15, line 9, is also introduced with little explanation.

We have been pragmatic and removed it in favour of "difficulties arising in..."

6) One typographical error is noted on page 13, the word consultation may require an 's' - just before

the insert box 1 caption.

Corrected

7) I think the paper engages with the contemporary complexity of consultations in a useful way. In doing so it has introduced a number of theoretical approaches and lenses to the consultation, i.e. case study methodology, 'telling case study', theme-oriented discourse analysis, linguistic ethnography, social constructionist perspectives, Conversation Analysis, ethnographic observation etc. In addition many linguistic terms like 'text', 'dyad', 'voice', 'interactional structure' etc. I do not know how the plethora of terms and approaches might be simplified - maybe this is not needed. But I would encourage the authors to consider what this might mean for some readers.

Thank you for drawing our attention to this and we absolutely agree that there are inevitable difficulties in bringing new disciplinary lenses to an audience of readers that may be unfamiliar with the terminology. Some of the terms you mention (e.g. conversation analysis, 'telling case') we have described briefly in the main body of the text. We have now added a box which includes some further definitions. We have refrained from listing here ALL the terms that might possibly be unfamiliar (since it may become rather like a mini textbook of sociolinguistics!) but we do hope we have covered the most important (and least familiar) ones where failure to understand them might result in a very impoverished engagement with the material.

VERSION 2 – REVIEW

REVIEWER	Geraldine Leydon University of Southampton UK
REVIEW RETURNED	14-Aug-2014

REVIEWER	Dr Marie Manidis University of Technology Sydney, Australia
REVIEW RETURNED	21-Aug-2014

GENERAL COMMENTS	This is a timely and useful contribution to a deeper understanding of what transpires in complex consultations and how GPs and medical educators might better address these complexities.
---

REVIEWER	Fiona Stevenson UCL, UK
REVIEW RETURNED	01-Sep-2014

GENERAL COMMENTS	I am now happy with the paper
-------------------------------